# Cervical Vertebral Stenotic Myelopathy in a Nelore Calf

**DOI:** 10.3390/vetsci9120699

**Published:** 2022-12-16

**Authors:** Mariana de Oliveira Bonow, José Renato Junqueira Borges, Isabel Luana de Macêdo, Davi Emanuel Ribeiro de Sousa, João Marcelo Azevedo de Paula Antunes, Márcio Botelho de Castro, Antonio Raphael Teixeira-Neto, Benito Soto-Blanco, Antonio Carlos Lopes Câmara

**Affiliations:** 1Large Animal Veterinary Teaching Hospital, College of Agronomy and Veterinary Medicine, Universidade de Brasília, Galpão 4, Granja do Torto, Brasília 70910-900, DF, Brazil; 2Veterinary Pathology Laboratory, College of Agronomy and Veterinary Medicine, Universidade de Brasília, Via L4 Norte, Campus Universitário Darcy Ribeiro, Asa Norte, Brasília 70910-900, DF, Brazil; 3Veterinary Hospital, Universidade Federal Rural do Semi-Árido, BR 110 Km 47 n 572, Mossoró 59625-900, RN, Brazil; 4Department of Veterinary Clinics and Surgery, Veterinary College, Universidade Federal de Minas Gerais, Avenida Presidente Antônio Carlos 6627, Belo Horizonte 31270-901, MG, Brazil

**Keywords:** compressive myelopathy, livestock, spinal cord disease, vertebral anomalies

## Abstract

**Simple Summary:**

This paper reports a case of cervical vertebral stenotic myelopathy (CVSM) affecting a Nelore (*Bos taurus indicus*) calf for the first time. CVSM is a result of vertebral malformation that causes compressive injury in the spinal cord. The clinical signs include weakness and ataxia. The cause of CVSM has not been well-determined, but inheritance, nutrition, and environmental factors have been proposed. The scarce information in the literature suggests that CVSM is underdiagnosed or under-reported in cattle.

**Abstract:**

This paper aims to report clinical, laboratory, radiographic, and pathological features in a case of cervical vertebral stenotic myelopathy (CVSM) affecting a 4-month-old Nelore calf for the first time. During physical examination, the calf could stand if assisted when lifting by the tail but fallen to the ground when trying to walk. Attempts to flex and extend the neck to the right side failed. Radiographs findings consisted of reduced intervertebral spaces, and misalignments between the endplates, more evident between the C3 and C4 vertebrae, resulting in narrowing of the spinal canal and compression of the spinal cord. Grossly, C4 showed cranial articular surface malformation, abnormal metaphyseal growth plate development, reduced vertebral body size and deformity. Histologically, C4 showed an abnormal vertebral bone development characterized by moderate replacement of trabecular bone by fibrous tissues, multifocal areas of dystrophic hyaline cartilage development, and cartilaginous growth failure along the metaphyseal growth plate. Cervical spinal cord within the stenotic vertebral canal showed swollen neurons with central chromatolysis, areas of Wallerian degeneration, and necrotic debris. In contrast with the well-known Wobbler syndrome in horses, the etiology of CVSM in cattle remains undetermined, and further genetic and pathological studies must be conducted to elucidate it.

## 1. Introduction

Spinal cord disease is relatively common in large animal practices. Wobbler syndrome is an old term used to designate cervical vertebral stenotic myelopathy (CVSM) in the veterinary literature. CVSM comprises clinical signs, such as weakness and ataxia, as a result of vertebral malformation and malalignment, cervical vertebral canal stenosis, and compressive injury in the spinal cord [1,2,3]. As a cause of noninfectious spinal cord ataxia, CVSM has been well characterized in horses and classified into type I, II and III, which affect sucklings and weanlings foals to 1–3-year-old horses with some differences in clinical and pathological presentations [2].

The etiology and pathogenesis of CVSM have not been well-determined, but inheritance, nutrition, and environmental factors have been proposed [1,2,3,4]. The paucity of information in the literature suggests that CVSM in cattle is underdiagnosed or under-reported [4]. Therefore, this paper aims to report clinical, laboratory, radiographic, and pathological features in a case of CVSM affecting a Nelore (*Bos taurus indicus*) calf for the first time.

## 2. Material and Methods

A 4-month-old, 140-kg Nelore calf was referred to the Large Animal Veterinary Teaching Hospital, University of Brasilia, Brazil, for history taking and systematic clinical examination [5]. Blood samples were collected through jugular venipuncture for hematology (complete blood count and fibrinogen determination) and serum biochemistry (aspartate aminotransferase, gamma-glutamyl transferase, urea, and creatinine levels) by routine laboratory methods [6]. Cerebrospinal fluid (CSF) was collected in the atlanto-occipital space for biochemical, cytological, and microbiological analysis [7]. Digital radiographic examinations were performed laterally on cervical region after sedation (0.05 mg/kg of xylazine).

The calf was humanely euthanized (0.1 mg/kg of xylazine and 1.5 g of thiopental intravenously followed by an intrathecal injection of 30 mL lidocaine) [8] and subjected to necroscopic examination. During necropsy, C3–C4 cervical spinal cord and vertebral samples were collected and fixed in a 10% buffered formalin solution (pH 7.0). After fixation, vertebral samples were decalcified with a 5% nitric acid solution. All tissue samples were embedded in paraffin, and histological sections were stained with hematoxylin and eosin (H&E) for microscopic evaluation. Histological sections from affected areas of cervical spinal cord were also stained with Luxol fast blue-cresyl violet stain (LFB).

## 3. Results

A 4-month-old, 140-kg Nelore calf was referred for physical examination due to the inability to stand after clinical evolution of 15 days. Initially, the owner stated that the calf dragged the claws of the hind limbs, evolving to paresis. The calf’s physical examination showed sternal recumbence, mild dehydration (assessed by skin turgor), a rectal temperature of 38.3 °C (100.9 °F), and ruminal and intestinal hypomotility. Additionally, the calf could stand if assisted when lifting by the tail, but fell to the ground when trying to walk, presenting symmetric hindlimbs ataxia. Attempts to flex and extend the neck to the right side failed. At the neurological evaluation, mentation and sensory and motor reflexes on all limbs and cranial nerves were unremarkable, but the tail and anal reflexes were reduced.

Haematological examination revealed leukocytosis (19.2 × 10^3^ leukocytes/μL; reference range: 4–12 × 10^3^ leukocytes/μL) by lymphocytosis (10.1 × 10^3^ lymphocytes/μL; reference range: 2.5–7.5 × 10^3^ lymphocytes/μL) and neutrophilia (7.6 × 10^3^ neutrophils/μL; reference range: 0.6–4.0 × 10^3^ neutrophils/μL). Biochemical abnormalities included hypoproteinemia (5.5 g/L; reference range: 6.74–7.46 g/L) by hypoalbuminemia (2.64 g/L; reference range: 3.0–3.55 g/L) and hypoglobulinemia (2.76 g/L; reference range: 3.0–3.48 g/L) [9]. Cerebrospinal fluid (CSF) analysis was unremarkable [7]. A CSF sample was referred for microbiological assay and tested negative. Lateral cervical digital radiographs identified the main finding of a cervical spondylopathy between the C3 and C4 vertebrae, causing stenosis of this vertebral canal due to misaligned vertebral bodies and abnormal endplates (C3 to C6) (Figure 1A). Dorsal dislocation of the C3 vertebral body was evident in association with near-total obliteration of its intervertebral foramen. The laminae of the C4, C5, and C6 vertebral arches appear to be small and abnormal, which alters the size and radiopacity of their intervertebral foramina. In the C2 to C6 articular processes, osteoarthrosis and bone proliferation are observed in the spinous processes. Ventrally to C4, bone proliferations were also noted. Spinal cord compression at the C3–C4 site was highly suspicious due to radiographic findings.

A presumptive diagnosis of CVSM causing spinal cord compression at C3 and C4 vertebrae was proposed, and, due to financial constraints, the owner elected euthanasia, and a necropsy was performed. Gross evaluation evidenced a ventral-dorsal misalignment between C3 and C4 vertebrae. The fourth cervical vertebra showed cranial articular surface malformation, abnormal metaphyseal growth plate development, and reduced vertebral body size and deformity. The medullary canal was also narrowed with focal compression of the cervical spinal cord (Figure 1B).

At microscopic evaluation, C4 showed an abnormal vertebral metaphyseal bone development characterized by islands of dystrophic hyaline cartilage surrounded by trabecular bone poorly mineralized (Figure 2A). Additionally, mild multifocal interstitial sclerosis of hyaline cartilage, multifocal areas of cartilaginous growth failure along the metaphyseal growth plate characterized by folding, irregular thickness, and prominent tongues of cartilage toward the metaphysis (osteochondrosis) were also observed (Figure 2B). Microscopical changes were not relevant in C3. The cervical spinal cord within the stenotic vertebral canal showed some swollen neurons with central chromatolysis. Additionally, the surrounding white matter evidenced multifocal to coalescent areas of Wallerian degeneration (Figure 3A) at the dorsal and lateral funicles, several axonal spheroids (Figure 3B), macrophages within digestion chambers, and necrotic debris. Histological sections from the affected cervical spinal cord stained with LFB showed bilateral symmetrical lesions, most on myelinated fibers within sensitive ascending dorsal and lateral bundles (Figure 4A,B), and motor descending lateral bundles (Figure 4B).

## 4. Discussion

Spinal cord compressive injuries have been classified as intramedullary, such as hemorrhages, neoplasms, and chronic expansile inflammatory disease, or extramedullary, as observed in the intervertebral disk herniation, CVSM, vertebral fracture, and luxation, neoplasms of the meninges, nerve sheath tumors, and, finally, vertebral bone developmental anomalies (vertebral deformities, scoliosis, lordosis, and kyphosis) [10]. Congenital anomalies of vertebrae have been reported in livestock, such as spina bifida, myelomeningocele, hemivertebrae, Arnold-Chiari malformation, complex vertebral malformation, brachyspina syndrome [1,11], and spinal stenosis affecting beef calves [4]. Congenital cervical vertebral malformation has been uncommonly reported in ruminants [12], especially CVSM causing clinical signs of spinal cord disease [13].

Spinal cord disease may cause a number of clinical signs (gait deficits resulting from weakness, ataxia or dysmetria, reflex abnormalities, and recumbency) depending on the location and severity of the lesion [11,14]. In the Nelore calf, the initial signs observed by the owner were hind limbs with claws dragging, reflecting an evident flexor weakness [1]. In the physical examination, sensory and motor reflexes were unremarkable in all limbs, but when the calf was assisted to stand, symmetric hindlimbs ataxia (proprioception deficits) and weakness were detected. These are considered characteristic findings of CVSM and reflect the severity of the cervical spinal cord injury in the present case. These signs are related to injuries in the white matter of the cervical spinal cord, mainly affecting the ascending general proprioception (causing ataxia) and the descending upper motor neurons (causing paresis) pathways [1,2,15]. Several *Bos taurus* calves with thoracic vertebral stenosis also showed similar clinical signs [4]. Injuries affecting the C1–C5 segment may present increased innate reflexes and decreased conscious sensation and proprioception in the limbs [14].

Although no significant changes were observed in the CSF analysis, leukocytosis may be related to spinal cord compression injuries [5], and serum protein changes could be due to a low feed intake caused by the calf’s inability to stand. In the Nelore calf, as reported in horses, plain lateral radiographs revealed some CVSM characteristic bony malformations, such as abnormal ossification of the articular processes, malalignment between adjacent vertebrae, and degenerative joint disease of the articular processes [3]. As previously reported, plain radiographs are relatively accurate at predicting a diagnosis of CVSM but should not be used to predict the site of compression due to unacceptable sensitivity and specificity [2]. Furthermore, myelography, computed tomography, epiduroscopy, and myeloscopy are considered more accurate for surgical planning and predicting spinal cord compression sites in horses. Surgical management of horses with spinal cord compression involves ventral cervical stabilization using locking compression plates, a polyaxial pedicle screw, a stainless-steel circular implant (Bagby basket), or a Seattle Slew (kerf-cut cylinder) implant [3].

Cervical vertebral malformation-malarticulation observed in the Nelore calf are pathological hallmarks of CVSM in ruminants [4,13,15] and horses [1,2,3]. As observed in the present case, gross and histological changes such as abnormal metaphyseal growth plate development and disruption, variable degree of vertebral anatomical deformations, and vertebral canal stenosis are the main pathological changes observed in affected calves [4]. In horses, abnormal bone and cartilage maturation have manifested as osteosclerosis and osteochondrosis in cases of CVSM [1,16], as also evidenced in the Nelore calf.

As a consequence of vertebral canal narrowing, severe spinal cord compressive injuries arose in the Nelore calf, such as Wallerian degeneration, axonal spheroids, and neuronal chromatolysis, as previously reported in other calves with thoracic vertebral stenosis [4]. Therefore, compressive spinal cord changes detected can give reasons for the calf’s proprioception deficits and the inability to stand. Differential diagnoses of CVSM include spinal cord and vertebral trauma and fractures, atlantoaxial subluxation, vertebral abscess, cervicothoracic subluxation, degenerative disc disease, and discospondylitis [1,13].

A few reports have failed to demonstrate an etiology for CVSM in ruminants [4,12,13], but some risk factors for the abnormal articular facets development and metaphyseal growth have been proposed in horses, such as inheritance and genetic predisposition, rapid growth, maternal nutrition, calcium/phosphorus imbalance, low dietary copper, and high dietary zinc [2,3]. A developmental theory for cervical vertebral malformations was proposed as a consequence of anomalous bone and cartilage morphogenesis and maturation in horses [1,16]. In contrast, a biomechanical theory has put forward an idea of abnormal mechanical forces and stresses on the cervical column in the pathogenesis of anatomical vertebral changes and stenosis of the vertebral canal [17]. In the present case, the beef cattle herd was raised extensively on native pastures (Brazilian cerrado), and only a single calf presented CVSM. Therefore, an etiology or risk factor for CVSM could not be determined in the Nelore calf. To the best of our knowledge, this is the first report of CVSM affecting cattle worldwide. In contrast to the well-known wobbler syndrome in horses, CVSM in cattle remains undetermined, and further genetic and pathological studies must be conducted to elucidate it.

## 5. Conclusions

CVSM was diagnosed in a 4-month-old Nelore calf for the first time. C4 showed cranial articular surface malformation, abnormal metaphyseal growth plate development, reduced vertebral body size, and deformity. Reduced intervertebral spaces and misalignments between the endplates, more evident between the C3 and C4 vertebrae, resulted in narrowing of the spinal canal and compression of the spinal cord. Cervical spinal cord within the stenotic vertebral canal showed swollen neurons with central chromatolysis, areas of Wallerian degeneration, and necrotic debris. In contrast with the well-known Wobbler syndrome in horses, the etiology of CVSM in cattle remains undetermined, and further genetic and pathological studies must be conducted to elucidate it.

## Figures and Tables

**Figure 1 vetsci-09-00699-f001:**
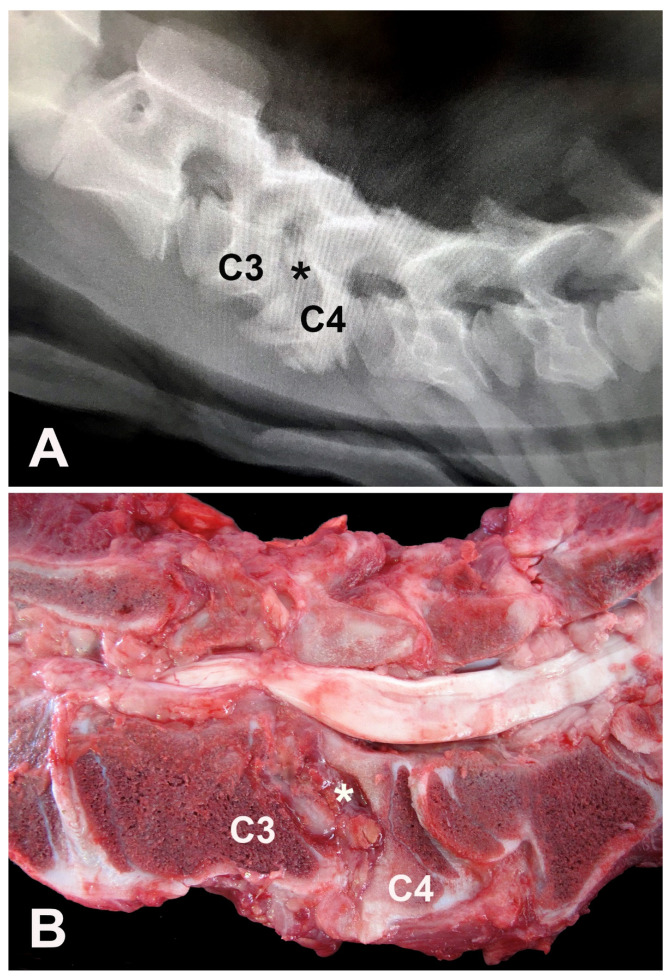
Four-month-old Nelore calf. (**A**) Latero-lateral digital radiograph of the cervical vertebral column showing intervertebral foramina with altered radiopacity, reduced intervertebral spaces, and misalignments between the endplates, more evident between the C3 and C4 vertebrae (asterisk) with material displacing the spinal cord ventri-dorsally and almost completely obliterating the corresponding intervertebral foramen, resulting in narrowing of the spinal canal and compression of the spinal cord. (**B**) Cervical vertebrae, longitudinal section, gross aspect. C3–C4 vertebrae misalignment (asterisk), reduced C4 vertebral body size and deformity, and focal compression of the cervical spinal cord.

**Figure 2 vetsci-09-00699-f002:**
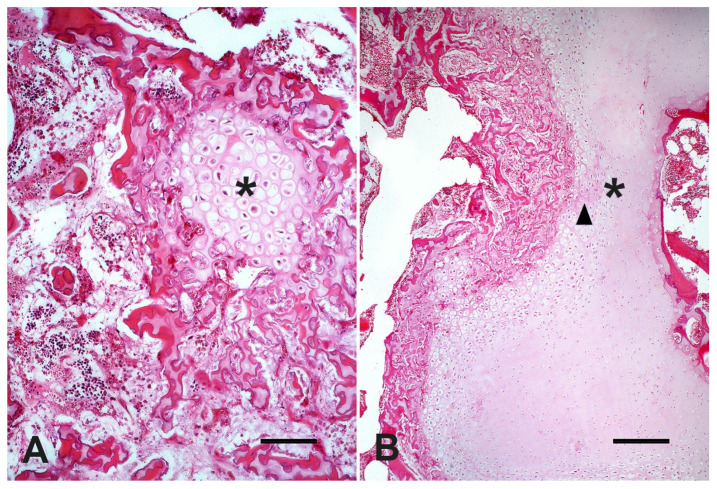
Cranial metaphyseal bone of the C4 vertebra of a four-month-old Nelore calf. (**A**) Abnormal area of dystrophic hyaline cartilage development (asterisk) (H&E, bar = 100 µm). (**B**) Cartilaginous growth failure (asterisk) and interstitial sclerosis of hyaline cartilage (arrowhead) (H&E, bar = 250 µm).

**Figure 3 vetsci-09-00699-f003:**
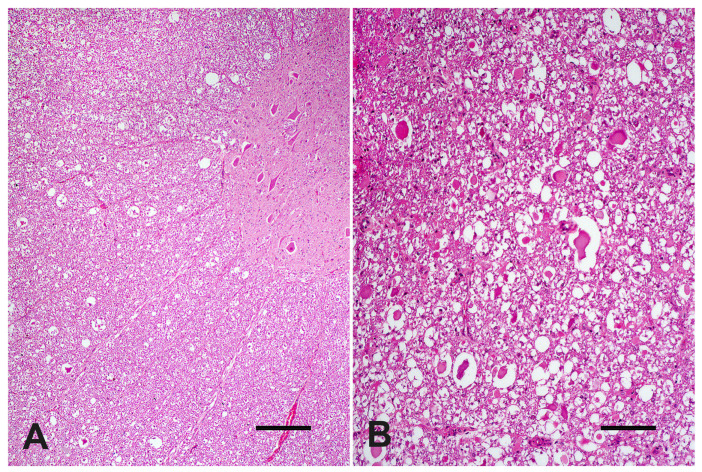
White matter of C3–C4 segment of the spinal cord of a four-month-old Nelore calf., (**A**) Multifocal areas of Wallerian degeneration (H&E, bar = 250 µm). (**B**) A severely affected area close to the spinal cord surface with several axonal spheroids (H&E, bar = 100 µm).

**Figure 4 vetsci-09-00699-f004:**
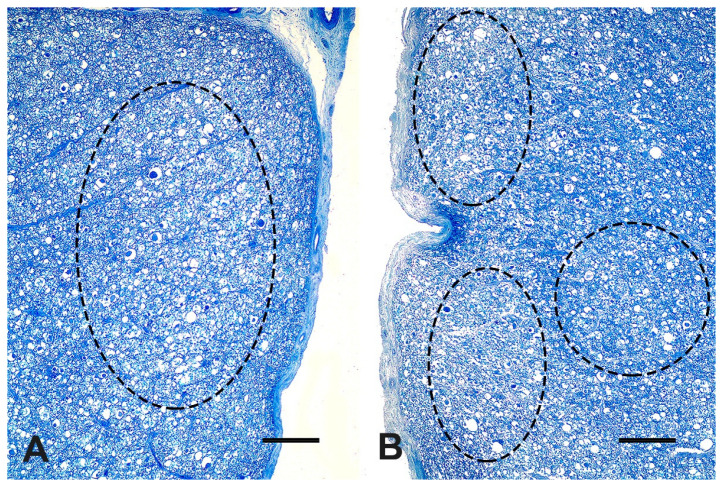
White matter of C3–C4 segment of the spinal cord of a four-month-old Nelore calf. (**A**) Left dorsal funicle, sensitive ascending bundles. Dilated myelin sheaths and multifocal axonal spheroids (Luxol fast blue-cresyl violet stain—LFB, bar = 250 µm). (**B**) Left lateral funicle, sensitive ascending bundles (ellipses), and motor descending bundles (circle). Wallerian degeneration and numerous axonal spheroids (LFB, bar = 250 µm).

## Data Availability

Not applicable.

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
