# Peer review of "Cervical Vertebral Stenotic Myelopathy in a Nelore Calf"

_vetsci, 2022, doi:10.3390/vetsci9120699_

Round 1
Author Response
Review of: Cervical Vertebral Stenotic Myelopathy in a Nelore Calf
Mariana de Oliveira Bonow1 , José Renato Junqueira Borges1 , Isabel Luana de Mâcedo2 , Davi Emanuel Ribeiro de 3 Sousa2 , João Marcelo Azevedo de Paula Antunes3 , Márcio Botelho de Castro2 , Antonio Raphael Teixeira-Neto1 , Be- 4 nito Soto-Blanco4,* and Antonio Carlos Lopes Câmara1
- In this case report a very rare vertebral malformation (CVSM cervical vertetbral stenothic myelopathy) in a calf (Nelore) is described.
- The topic is very relevant because it has not been described yet and can help practitioners with finding the diagnosis.
- What does it add to the subject area compared with other published material?
It first describes a very rare vertebral malformation (CVSM cervical vetretbral stenothic myelopathy) in a calf (Nelore).
- What specific improvements could the authors consider regarding the methodology?
The case report provides enough background information and describes the case in detail.
- Are the conclusions consistent with the evidence and arguments presented and do they address the main question posed?
Yes
- Are the references appropriate?
Yes
R: We appreciate all the considerations and suggestions for manuscript improvement. We are responding to each issue in turn and in detail below.
- Please include any additional comments on the tables and figures.
Figure 1: B: the asterisk is missing
R: We corrected Figure 1 as observed.
- Points that need to be addressed by the authors:
Line 57: what exactly do you mean with: “the calf emulated the hind legs”? Should it maybe be: the calf dragged the hind limbs?
R: The reviewer is correct. This was a translation mistake. The sentence has corrected. Line 58.
Line 134: instead of: ataxia was evidenced – better “was detected”
R: The entire paragraph was restructured due to observations made by Reviewer 2. We changed the sentence as suggested. Line 157.
Line 165: I would not say “it gives ground for the proprioceptive deficits” but better “it gives reasons for…” or this might be the reason…
R: We changed the manuscript as suggested. Line 198.

Reviewer 2 Report
Dear authors, the case report submitted is really the first one described in bovine species, in accordance with the international scientific literature. The Case report is noteworthy for its uniqueness in cattle but needs more histopathological investigations for making it eligible for publication in the Veterinary Science, journal renowned for its high scientific impact in the international reference field. For increase the scientific level would be necessary to carry out a histochemical evaluation of the myelin in spinal cord formalin-fixed and paraffin-embedded tissue: Luxol fast blue - cresyl violet stain. The presence of axonal spheroids in the lateral and dorsal funiculi are a very important observation but more detailed information about unilateral or bilateral and/or symmetrical or asymmetrical lesions need as well as other ones on myelin status. Myelin investigation is necessary to first identify which type of axonal fiber are involved (myelinated or unmyelinated or both) and contextualize them in the spinal tracts (ascending or descending bundles) most involved in order to give an anatomical concordance between clinical signs and nerve lesions in the chapter "Discussion".
Line 55 - replace "evaluation" with "examination";
Line 94 - mention in bibliography "AVMA Guidelines for the Euthanasia Of Animals: 2020 Edition2 - https://www.avma.org/sites/default/files/2020-02/Guidelines-on-Euthanasia-2020.pdf
Author Response
Comments and Suggestions for Authors
Dear authors, the case report submitted is really the first one described in bovine species, in accordance with the international scientific literature. The Case report is noteworthy for its uniqueness in cattle but needs more histopathological investigations for making it eligible for publication in the Veterinary Science, journal renowned for its high scientific impact in the international reference field. For increase the scientific level would be necessary to carry out a histochemical evaluation of the myelin in spinal cord formalin-fixed and paraffin-embedded tissue: Luxol fast blue - cresyl violet stain. The presence of axonal spheroids in the lateral and dorsal funiculi are a very important observation but more detailed information about unilateral or bilateral and/or symmetrical or asymmetrical lesions need as well as other ones on myelin status. Myelin investigation is necessary to first identify which type of axonal fiber are involved (myelinated or unmyelinated or both) and contextualize them in the spinal tracts (ascending or descending bundles) most involved in order to give an anatomical concordance between clinical signs and nerve lesions in the chapter "Discussion".
R: We appreciate all the considerations and suggestions for manuscript improvement. We changed the manuscript as suggested. Lines 118-121 and included Figures 4A and 4B to fix this issues as suggested. An anatomical concordance between clinical signs and nerve lesions was added Lines 151-164.
Line 55 - replace "evaluation" with "examination";
R: We changed the manuscript as suggested. Line 55
Line 94 - mention in bibliography "AVMA Guidelines for the Euthanasia Of Animals: 2020 Edition2 - https://www.avma.org/sites/default/files/2020-02/Guidelines-on-Euthanasia-2020.pdf
R: The reference was added. Line 97. Lines 247-249.

Author Response
This manuscript written by Bonow et al describes the clinical, laboratory, gross and microscopic features of cervical vertebral stenotic myelopathy in a Nelore calf. The manuscript is interesting since CVSM is poorly described in ruminants. However, the authors need to improve the pathological description and interpretation of the findings (see below) including the link between the type of misalignment-topographic areas of the Wallerian degeneration in the spinal cord-clinical signs.
R: We appreciate all the considerations and suggestions for manuscript improvement. We are responding to each issue in turn and in detail below.
L20. The scarce information…
R: We corrected the manuscript as suggested. Line 20
L44-47. If you say three different types of CVSM, at least add the names of the classification.
R: We reformulated the manuscript as suggested. Line 45
L55. Where was the calf referred to? The sentence misses the place where the animal was referred.
R: We reformulated the manuscript to fix this issue. Lines 55-56.
L84. Figure 1. Identify the type of vertebrae (C1, C2, etc.) in the radiograph and gross images.
R: We identified the type of vertebrae in the radiograph and gross images as suggested. Figure 1
L90. The asterisk is not in the figure 1B.
R: We reformulated Figure 1B to fix this issue.
L89-L91. The pathological description of the vertebrae is poor. I can see pathological thickening of the bone in the vertebrae (sclerosis-see asterisk in the below image) in the areas of compression, but there are more details that have been omitted. Add type of vertebrae (C1, C2, etc.) for better description.
R: The pathological description of the vertebrae was improved as suggested. Lines 106-113.
L106. In the paragraph, the authors mention “fibrous tissue”, which is not showed in Figure 2A.
R: We reformulated most histological descriptions and Figure 2A to fix this issue. Lines 106-113.
L114. Figure 2A, was the picture taken near the growth plate? I would hesitate to say that it is hyaline cartilage because it might be lack of endochondral ossification of the growth plate with extension (tongues) of the hypertrophic zone. Figure 2B, add symbols in the figure since it is difficult to orient the readers, is the tissue on the left side the intervertebral disc?
R: We appreciate your observations very much. We reformulated most of the histological description and legend of Figure 2A to fix this issue. Lines 106-113.

Round 2
Reviewer 2 Report
Dear authors, the paper is ready to print.
Please refine one word and and type-write error:
1) line 83 change Hematological in Haematological;
2) line 133 "andinterstitial" put a space between the words "and interstitial"